# Low-Volume Metastases in Apparent Early-Stage Endometrial Cancer: Prevalence, Clinical Significance, and Future Perspectives

**DOI:** 10.3390/cancers16071338

**Published:** 2024-03-29

**Authors:** Diletta Fumagalli, Luigi A. De Vitis, Giuseppe Caruso, Tommaso Occhiali, Emilia Palmieri, Benedetto E. Guillot, Giulia Pappalettera, Carrie L. Langstraat, Gretchen E. Glaser, Evelyn A. Reynolds, Robert Fruscio, Fabio Landoni, Andrea Mariani, Tommaso Grassi

**Affiliations:** 1Department of Obstetrics and Gynecology, Division of Gynecologic Oncology, Mayo Clinic, Rochester, MN 55905, USA; fumagalli.diletta@mayo.edu (D.F.); devitis.luigiantonio@mayo.edu (L.A.D.V.); caruso.giuseppe@mayo.edu (G.C.); occhiali.tommaso@mayo.edu (T.O.); palmieri.emilia@mayo.edu (E.P.); guillot.benedetto@mayo.edu (B.E.G.); pappalettera.giulia@mayo.edu (G.P.); langstraat.carrie@mayo.edu (C.L.L.); glaser.gretchen@mayo.edu (G.E.G.); reynolds.evelyn@mayo.edu (E.A.R.); mariani.andrea@mayo.edu (A.M.); 2Department of Medicine and Surgery, University of Milan-Bicocca, 20126 Milan, Italy; robert.fruscio@unimib.it (R.F.); fabio.landoni@unimib.it (F.L.); 3Department of Gynecology, European Institute of Oncology (IEO) IRCCS, 20141 Milan, Italy; 4Clinic of Obstetrics and Gynecology, Santa Maria della Misericordia University Hospital, Azienda Sanitaria Universitaria Friuli Centrale, 33100 Udine, Italy; 5Gynecologic Oncology Unit, Department of Women, Children and Public Health Sciences, , Fondazione Policlinico Universitario Agostino Gemelli IRCCS, 00136 Roma, Italy; 6Division of Gynecologic Surgery, IRCCS Fondazione San Gerardo dei Tintori, 20900 Monza, Italy

**Keywords:** endometrial cancer, sentinel node mapping, ultrastaging, low-volume metastasis, isolated tumor cells, micrometastasis

## Abstract

**Simple Summary:**

In apparent early-stage endometrial cancer, sentinel lymph node sampling and ultrastaging have allowed for the detection of micrometastases and isolated tumor cells grouped as low-volume metastases. The prevalence and clinical significance of low-volume metastases are described and discussed, with a final focus on areas of potential future investigation.

**Abstract:**

Endometrial cancer (EC) is the most diagnosed gynecologic malignancy, and its incidence and mortality are increasing. The prognosis is highly dependent on the disease spread. Surgical staging includes retroperitoneal evaluation to detect potential lymph node metastases. In recent years, systematic lymphadenectomy has been replaced by sentinel lymph node (SLN) biopsy and ultrastaging, allowing for the detection of macrometastases, micrometastases, and isolated tumor cells (ITCs). Micrometastases and ITCs have been grouped as low-volume metastases (LVM). The reported prevalence of LVM in studies enrolling more than one thousand patients with apparent early-stage EC ranges from 1.9% to 10.2%. Different rates of LVM are observed when patients are stratified according to disease characteristics and their risk of recurrence. Patients with EC at low risk for recurrence have low rates of LVM, while intermediate- and high-risk patients have a higher likelihood of being diagnosed with nodal metastases, including LVM. Macro- and micrometastases increase the risk of recurrence and cause upstaging, while the clinical significance of ITCs is still uncertain. A recent meta-analysis found that patients with LVM have a higher relative risk of recurrence [1.34 (95% CI: 1.07–1.67)], regardless of adjuvant treatment. In a retrospective study on patients with low-risk EC and no adjuvant treatment, those with ITCs had worse recurrence-free survival compared to node-negative patients (85.1%; CI 95% 73.8–98.2 versus 90.2%; CI 95% 84.9–95.8). However, a difference was no longer observed after the exclusion of cases with lymphovascular space invasion. There is no consensus on adjuvant treatment in ITC patients at otherwise low risk, and their recurrence rate is low. Multi-institutional, prospective studies are warranted to evaluate the clinical significance of ITCs in low-risk patients. Further stratification of patients, considering histopathological and molecular features of the disease, may clarify the role of LVM and especially ITCs in specific contexts.

## 1. Introduction

Endometrial cancer (EC) is the fourth most commonly diagnosed cancer in women, with an estimated 66,880 new cases and 13,240 deaths in the US in 2024 [1]. Patients with early-stage disease (2009 International Federation of Gynecology and Obstetrics (FIGO) stage I–II) [2] have a good prognosis, with five-year overall survival rates exceeding 80%, while patients with advanced-stage disease (2009 FIGO stage III–IV) [2] have poorer outcomes [3]. If the disease has spread to the pelvic lymph nodes, the 5-year survival rate is 58% (2009 FIGO stage IIIC1) [3]. The anatomical extent of the disease, including metastases to the lymph nodes, is an important prognostic factor. In addition, the histopathological subtype [4] and the molecular features of the disease [5] have also been described as independent predictors of prognosis. In the 2023 FIGO staging, the presence of an aggressive histopathological subtype also leads to upstaging when the disease is confined to the uterus if there is myometrial invasion [6]. Aggressive histopathological subtypes include high-grade endometrioid cancers, serous, clear cell, undifferentiated, mixed, mesonephric-like, gastrointestinal mucinous type carcinomas, and carcinosarcomas [6]; it is well known that in non-endometrioid disease survival is significantly reduced [7]. In addition to traditional histopathologic findings and disease spread, The Cancer Genome Atlas (TCGA) [5] identified four molecular EC subgroups with distinct prognostic characteristics: copy-number high with poor prognosis, copy-number low and mismatch-repair deficient with intermediate prognosis, and POLE-mutated with an overall good prognosis. This classification relied on expensive analyses of fresh frozen samples, but new systems based on the same categories as identified by TCGA—like the Proactive Molecular Risk Classifier for Endometrial Cancer (ProMisE)—proposed more pragmatic methods that can be leveraged in clinical practice [8,9]. The updated molecular classification includes four groups: p53-abnormal (instead of copy-number high), non-specific molecular profile (instead of copy-number low), mismatch-repair deficient, and POLE-mutated [9]. This classification has been included in the latest ESGO/ESTRO/ESP guidelines [10] and 2023 FIGO staging [6]. 

The staging system of EC has been evolving over time. In 1988, the International Federation of Gynecology and Obstetrics (FIGO) changed the staging system for endometrial cancer from clinical to surgical staging, which included pelvic and para-aortic lymphadenectomy [11]. This change was made because complete surgical staging can detect occult nodal disease and guide adjuvant treatment. The importance of surgical staging was further confirmed in the 2009 revised FIGO staging system [12], and the latest version was published in 2023 [6]. However, in current surgical practice, not all surgeons perform extensive retroperitoneal staging [13,14,15,16]. The ASTEC (A Study in the Treatment of Endometrial Cancer) study group [17] and Benedetti Panici et al. [18] demonstrated that systematic pelvic lymphadenectomy does not improve survival rates and may increase postoperative morbidity. However, the impact of pelvic and para-aortic lymphadenectomy on overall survival in EC at high risk of recurrence is still debated. The ongoing Endometrial Cancer Lymphadenectomy (ECLAT) Trial aims to provide further insight into this matter [NCT03438474] [19].

The sentinel lymph node (SLN) technique in EC was first explored in 1996 by Burke et al. [20] as a less invasive but equally safe surgical procedure to perform retroperitoneal staging and further validated by subsequent studies [21,22,23,24,25,26]. Current international guidelines recommend retroperitoneal evaluation during surgery for EC to detect potential nodal involvement. According to the 2021 ESGO/ESTRO/ESP guidelines [10], a negative sentinel node can confirm N0 status. This is supported by both retrospective and prospective cohort studies, which have shown a high sensitivity and negative predictive value of SLN mapping [27,28,29,30,31,32,33]. This finding has also been confirmed for high-risk EC [25,26,34]. The National Comprehensive Cancer Network (NCCN) clinical practice guidelines recommend including SLN mapping as a staging procedure [35] because it may improve the detection of lymph node involvement in apparent early-stage disease [36,37]. Extensive histopathologic analysis of nodal tissue, known as ultrastaging, has been successfully applied to nodal tissue biopsied using SLN mapping techniques. This procedure has led to the discovery of small amounts of cancerous tissue in the sentinel nodes, termed low-volume metastases (LVM). The prevalence and clinical significance of LVM detected in the sentinel nodes during EC staging are discussed in the present review.

### 1.1. Sentinel Node Mapping Technique: Site of Injection, Dye, and Sentinel Node Algorithm

Several injection techniques have been tested to maximize the detection of SLN mapping. Cervical injection with dye has been recognized as the most appropriate procedure [38,39]. Abu-Rustum et al. [38] described the technique to inject dye into the cervical stroma at the 3 and 9 o’clock positions in the operating room after examination under anesthesia (a cervical injection example is represented in Figure 1, panel 1).

A Society of Gynecologic Oncology literature review [40] by Holloway et al. reported that colorimetric lymphatic mapping using isosulfan blue or methylene blue as a single marker had an overall SLN detection rate of 71%, which increased to 86% with the combined use of blue dye and radiolabeled colloid technetium 99 (Tc99). The use of indocyanine green (ICG) fluorescence, although requiring a near-infrared camera for localization, showed a higher overall (95%) and bilateral (66%) detection rate [40]. This finding was also reported in a systematic review and meta-analysis by Burg et al. [41]. Compared to isosulfan blue, ICG with near-infrared fluorescence imaging had superior detection capability in identifying SLNs in women with cervical and uterine cancer [42]: in fact, 96% of SLNs were identified with ICG and 46% with isosulfan blue (difference of 50%, 95% CI 39–61; *p* < 0.0001) [42].

Barlin et al. [43] proposed an SLN mapping algorithm including peritoneal and serosal evaluation and washing, retroperitoneal evaluation including excision of all mapped SLNs, and removal of all suspicious nodes regardless of mapping. If no SLNs are marked in a hemipelvis, side-specific pelvic, common iliac, and interiliac lymph node dissection is performed. As suggested by the mapping algorithm, paraaortic lymphadenectomy should not be routinely performed but can be left to the surgeon’s discretion. When compared retrospectively with standard bilateral lymphadenectomy, this algorithm did not adversely affect the detection rate of positive lymph nodes [44].

### 1.2. Ultrastaging of the Sentinel Nodes

Traditional histologic examination of lymph nodes involves a cross-sectional cut along their longitudinal axis and hematoxylin and eosin staining of the two cut surfaces [45]. In the 1990s, studies on breast cancer [46] and melanoma [47] reported that improved pathologic evaluation of lymph nodes, including intensive analysis of multiple sections and immunohistochemistry—later referred to as ultrastaging—could detect metastases not previously identified by hematoxylin and eosin staining.

In 2001, Yabushita et al. [48] first reported immunostaining results for cytokeratin in 304 lymph nodes from 46 patients with endometrioid adenocarcinoma. In the following years, different ultrastaging protocols have been proposed. Major differences between protocols have been found in the method of macroscopic slicing, which can be longitudinal or “bread-loaf”, the interval used for serial sectioning, and the number of slides used for ultrastaging [45]. Ultrastaging of SLN has been proposed and validated in EC as these lymph nodes have the highest risk of involvement [49,50].

Although ultrastaging protocols vary between institutions, they typically include hematoxylin and eosin staining of one slide at each section level, with two unstained at the same level. Pankeratin immunohistochemistry is performed on one unstained slide at each level if staining is negative for metastasis. The Memorial Sloan Kettering SLN processing protocol includes the following: (1) hematoxylin/eosin staining; (2) if standard examination is negative, from each paraffin block two adjacent 5 µm sections at each of two levels, 50 µm apart, are cut longitudinally; and (3) at each level one slide is stained with hematoxylin and eosin and one with immunohistochemistry using anticytokeratin AE1:AE3 for a total of five slides per block [49]. At MD Anderson, two different protocols—an intensive method and an abbreviated version—were compared [51], and no difference in detecting nodal metastases was found. In the first protocol, five levels are stained with hematoxylin and eosin at 250 µm intervals, with two unstained slides at each level; then, pankeratin immunohistochemistry is performed on one slide in cases with negative hematoxylin and eosin examination. The second protocol involves staining with hematoxylin/eosin only on one level instead of five, with two accompanying unstained slides. The ultrastaging steps are shown schematically in Figure 1, panel 2.

Ultrastaging is performed after permanent sectioning of the SLN. A potential limitation of SLN ultrastaging on permanent sections, without frozen section analysis, is that in around 8% of cases, an empty node is found (i.e., a specimen that contains only fibroadipose and no lymphoid tissue at the pathologic evaluation) [52]. Casarin et al. proposed frozen-section analysis as a method to determine the accuracy of the SLN bioptic procedure, ensuring adequate nodal staging [52]. Bellaminutti et al. [53] also reported that the accuracy of the frozen section in detecting nodal metastases is 93%, with a positive predictive value of 100%. A systematic review and meta-analysis [54] also reported that intraoperative SLN analysis on frozen sections increased the SLN detection rates. However, the ESGO/ESTRO/ESP guidelines state that frozen section analysis should not replace final pathologic examination and ultrastaging [10].

Recently, new molecular techniques have been explored to detect nodal metastases in SLN. Nagai et al. [55] first reported on the effectiveness of a one-step nucleic acid amplification (OSNA) assay using cytokeratin 19 (CK19) mRNA for diagnosing LN metastasis in EC patients. Normal lymphoid tissue does not express CK19, but cancer cells and their metastases do. The authors found that the new diagnostic method had the same diagnostic capability as the frozen-section histology, with high concordance between the two methods (97.1%). The technique was later applied to 135 SLN (58 patients) and compared with ultrastaging, leading to the upstaging of 20.7% of cases in a comparative study by Kost’un et al. [56]. They reported a sensitivity of the OSNA assay of 90.9%, a specificity of 85.5%, and concordance of 85.9% with ultrastaging.

In 2023, a multicenter prospective study on the detection of nodal metastases in cervical and endometrial cancer patients, conducted in six institutions in Japan [57], showed that the diagnostic ability of the OSNA assay in detecting metastases was equivalent to that of histopathological examination. Specifically, the authors reported a concordance rate of 97.9% (95% CI: 96.1–99.1%), with a sensitivity and specificity of the OSNA assay of 91.8% (95% CI: 81.9–97.3%) and 98.9% (95% CI: 97.3–99.7%), respectively. Each lymph node (of 437 samples retrieved from 133 endometrial and cervical cancer patients who underwent systematic lymphadenectomy) was cut with 2 mm intervals along the short axis direction, and alternate sections were used for the OSNA assay and standard hematoxylin–eosin examination. Ultrastaging with anti-CK19 antibody immunostaining was not routinely performed. This procedure was performed in cases of non-concordance with the standard pathological examination and OSNA assay.

In conclusion, the OSNA assay could be a promising new method to diagnose nodal metastases in patients with EC. However, its cost-effectiveness compared to standard histopathologic examination and CK19 immunostaining, as well as broad applicability in standard clinical practice, has not yet been assessed.

## 2. Prevalence of Low-Volume Metastases

According to the American Joint Committee on Cancer (AJCC) definition [58], micrometastases are defined as microscopic clusters and single neoplastic cells where the greatest dimension is between 0.2 and 2 mm. In contrast, they are classified as ITCs when their greatest dimension is equal to or less than 0.2 mm (Figure 1, panel 3). Their prevalence—often reported grouped as LVM or as separate entities—has been extensively described in patients with EC, and in studies enrolling more than one thousand patients with apparent early-stage EC, it ranged from 1.9% to 10.2% [59,60]. In Table 1 and Table 2, the prevalence of LVM (either micrometastases or ITCs) is reported by dividing the studies according to the characteristics of the patient cohort.

In 2006, Ameczua et al. [61] examined the lymph nodes sampled from 76 patients with EC who had undergone pelvic and/or paraaortic nodal dissection with immunohistochemistry. Their lymph nodes were stored and reported as histologically negative for metastatic disease, and it was found, after ultrastaging, that fifteen patients (19.7%) had ITCs in their lymph nodes; six of them also had ITC-positive paraaortic nodes. Subsequent studies have focused on the prevalence of LVM in patients with apparent early-stage disease undergoing primary surgery for EC and SLN mapping.

**Table 1 cancers-16-01338-t001:** Prevalence of low-volume metastases in apparent early-stage EC.

First Author	Study Period	Country	Number of Patients	Patients with LVM (%)	Patients with MM (%)	Patients with ITCs (%)
Studies reporting data on prevalence of LVM, MM, and ITCs						
Ballester [31]	2007–2009	France	111	7 (6.3)	6 (5.4)	1 (0.9)
Holloway [37]	2006–2013	USA	119	22 (18.5)	10 (8.4)	12 (10.1)
Clinton [62]	2012–2015	USA	185	12 (6.5)	3 (1.6)	9 (4.9)
Euscher [51]	2007–2014	USA	178	22 (12.4)	14 (7.9)	8 (4.5)
Plante [63]	2010–2015	Canada	519	42 (8)	11 (2)	31 (6)
Rossi [25]	2012–2015	USA	293	19 (6.5)	9 (3.1)	10 (3.4)
Backes [64]	2013–2016	USA	184	11 (6)	2 (1.1)	9 (4.9)
Kennard [65]	2011–2016	USA	414	57 (13.7)	21 (5)	36 (8.7)
Martinelli [66]	2005–2019	Italy	221	14 (6.3)	6 (2.7)	8 (3.6)
Garcia Pineda [67]	2007–2016	Spain	230	14 (6.1)	8 (3.5)	6 (2.6)
Mueller [60]	2005–2018	USA	1044	106 (10.2)	45 (4.4)	61 (5.8)
Lavecchia [59]	2015–2019	Canada, Korea	1012	19 (1.9)	7 (0.7)	12 (1.2)
Buda [68]	2012–2020	Europe	1428	76 (5.3)	50 (3.5)	26 (1.8)
Total			5938	421 (7.1)	192 (3.2)	229 (3.9)
Studies reporting data on prevalence of LVM only						
Desai [69]	2011–2013	USA	103	5 (4.9)	NR	NR
Studies reporting data on prevalence of ITCs only						
Goebel [70]	2012–2016	USA	155	NR	NR	21 (13.5)
Matsuo [71]	2018	USA	6472	NR	NR	111 (1.7)
Mumford [72]	2017–2020	USA	848	NR	NR	33 (3.9)

Abbreviations: EC: endometrial carcinoma; LVM: low-volume metastases; MM: micrometastases; ITCs: isolated tumor cells; NR: not reported.

**Table 2 cancers-16-01338-t002:** Prevalence of low-volume metastases in low-, intermediate- or high-risk subgroups.

First Author	Study Period	Definition of Risk	Country	Number of Patients	Patients with LVM (%)	Patients with MM (%)	Patients with ITCs (%)
Raimond [73]	2000–2012	Apparent stage I, ESMO low- ^§^ and intermediate-risk ^¥^ EC	France	136	15 (11)	NR	NR
Todo [74]	1997–2004	Intermediate-risk EC: early-stage with any of the following: >50% MI, grade 3 disease, non-endometrioid, with cervical involvement, LVSI, PPC	Japan	61	9 (14.8)	3 (5)	6 (9.8)
Zahl-Eriksson [75]	2004–2013	Low-risk EC: MI < 50%, endometrioid, any grade	USA	642	22 (3.4)	2 (0.3)	20 (3.1)
Ducie [76]	2004–2013	Intermediate-risk EC: apparent early-stage with >50% MI, any grade, endometrioid; high-risk EC: serous, clear-cell	USA	202	20 (9.9)	8 (4)	12 (5.9)
Persson [33]	2014-2018	High-risk EC: apparent early stage with any of the following: endometrioid grade 3, non-endometrioid, >50% MI, CSI, non-diploid cytometry	Sweden	257	19 (7.4)	9 (3.5)	10 (3.9)
Bjornholt [77]	2017–2022	Apparent early-stage endometrioid EC, low-grade, any MI	Denmark	591	36 (5.7)	20 (3.1)	16 (2.6)
Burg [78]	2016–2021	Apparent early-stage, ESGO low- ^£^ and intermediate-risk ^¤^ EC	Netherlands	152	NR	NR	3 (2)

Abbreviations: EC: endometrial carcinoma; LVM: low-volume metastases; MM: micro-metastases; ITCs: isolated tumor cells; LVSI: lymphovascular space invasion; PPC: positive peritoneal cytology; MI: myometrial invasion; NR: not reported. ^§^ ESMO low-risk: endometrioid, grade 1–2, stage IA; ^¥^ ESMO Intermediate-risk: endometrioid, stage IA, grade 3, or stage IB grade 1 or 2; ^£^ ESGO low-risk: stage IA endometrioid, low-grade, LVSI negative or focal; ^¤^ ESGO intermediate-risk: stage IB endometrioid, low-grade, LVSI negative or focal OR stage IA endometrioid, high-grade, LVSI negative or focal OR stage IA non-endometrioid (serous, clear cell, undifferentiated carcinoma, carcinosarcoma, mixed) without myometrial invasion.

### 2.1. Low-Volume Metastases in Apparent Early-Stage Endometrial Cancer

In 2014, Desai et al. described the sentinel node mapping outcomes of 120 patients with apparent early-stage disease who underwent intraoperative lymphatic mapping using a methylene blue cervical injection followed by robotic SLN dissection [69]. Of 103 patients successfully mapped, 10 had positive nodes, and 5 of these had micrometastases (9.7 and 4.9%, respectively). Another study by Holloway et al. [37], comparing SLN mapping plus staging lymphadenectomy versus lymphadenectomy alone, found that among SLN-mapped patients (*n* = 119), 22 patients had low-volume disease, while 12 patients had ITCs only (18.5% and 10.1% of the population, respectively). A retrospective study published in 2017 by Plante et al. [63] found that among 519 patients with apparent early-stage disease who underwent complete pelvic lymph node dissection following SLN mapping, 42 patients had LVM in their lymph nodes (8.1%), of whom 31 (6%) had ITCs. The FIRES trial, a prospective, multicenter study by Rossi et al. [25]. comparing the accuracy of the SLN mapping technique with complete lymphadenectomy in apparent early-stage disease, reported that 19 patients out of 293 successfully mapped had LVM in their sentinel nodes—of which 9 had micrometastases, while 10 had ITCs (3.1% and 3.4%, respectively). Prevalence rates of micrometastases and ITCs in apparent early-stage disease have also been described by other authors [62,64,66,67,70,72] and are shown in Table 1.

A retrospective, single-center cohort study by Mueller et al. [60], published in 2020, and two multicenter studies [59,68], published in 2023, reported the prevalence of micrometastases and ITCs in patient cohorts comprising more than one thousand patients. Mueller et al. [60] reported the prevalence of ITCs in a cohort of 1044 patients who underwent primary surgical staging with successful bilateral SLN mapping at Memorial Sloan Kettering Cancer Center. Patients were further classified by grade, stage, and histology. They found that the presence of ITCs was directly related to the extent of myometrial invasion. The prevalence of ITCs in grade 1, endometrioid EC without myometrial invasion was less than 1% (2 patients out of 449), and no micrometastases or macrometastases were found in this group. In contrast, when the cancer spread involved the outer half of the myometrium, ITCs were reported in 19 grade 1 endometrioid EC cases out of 62 (31%), and micrometastases or macrometastases were found in 6 cases (10%). Of note, nodal involvement was most common in EC with non-endometrioid histology, even in the absence of myometrial invasion. ITCs, micrometastases, and macrometastases could be found in up to 10% of cases.

Buda et al. [68] also reported the prevalence of LVM in a multicenter, retrospective cohort of 1428 patients with apparent early-stage disease who underwent SLN mapping. LVM were present in 76 of 1387 successfully mapped patients, of which 50 were micrometastases (3.6%), and 26 were ITCs (1.9%). The retrospective, multicenter study published by Lavecchia et al. [59] reported the positive SLN rate in a cohort of 1041 patients who underwent SLN mapping, of whom 951 had either unilateral or bilateral successful mapping. Six patients had micrometastases, and eleven had ITCs in their SLNs (0.6% and 1.2%, respectively).

The prevalence of LVM has been investigated not only in a general cohort of patients with apparent early-stage disease but also in patient sub-cohorts with specific disease characteristics or that were selected after risk stratification. It should be noted that the definition of risk class is sometimes different between studies.

### 2.2. Prevalence of Low-Volume Metastases after Stratification of Patients in Low- and Intermediate-Risk Groups

A study by Raimond et al. published in 2014 reported a prevalence of micrometastases of 11% (15 patients out of a total of 136) in an apparent early-stage cohort of patients with EC defined as low risk (stage IA grade 1 or 2) or intermediate risk (stage IA grade 3 or stage IB grade 1 or 2) [73]. All patients underwent an SLN procedure followed by systematic pelvic lymphadenectomy. Zahl Eriksson et al. [75] reported a prevalence of LVM of 3.4% in a multicenter, retrospective study cohort of 642 patients with stage IA, endometrioid, and any grade EC who underwent SLN mapping for a total of 22 patients with LVM. Among these cases, 20 patients (3.1% of the total cohort) had ITCs. Bjornholt et al. [77] found a higher prevalence of LVM (6.1%) in a prospective cohort of 591 SLN-mapped patients with apparent early-stage, endometrioid, low-grade EC but a similar rate of ITCs (2.7%).

A retrospective analysis by Matsuo et al. [71] described the prevalence of ITCs and the characteristics of patients with ITCs in a population drawn from the US National Cancer Institute’s Surveillance, Epidemiology, and End Result Program (SEER). A total of 6472 patients with stage I EC who underwent primary hysterectomy and surgical nodal evaluation were included in the study. The authors reported a prevalence of ITCs of 1.7% (111 patients). Of note, 34 patients (38.2% of patients with ITCs, 0.5% of the total cohort) had low-risk features.

Todo et al. [74] reported the prevalence of ITCs in 61 patients with intermediate-risk EC who underwent primary surgery and pelvic lymphadenectomy alone (85.2%) or pelvic and para-aortic lymphadenectomy (14.8%). They included patients with apparent early-stage EC with at least one of the following characteristics: >50% myometrial invasion, grade 3 disease or non-endometrioid histology, cervical involvement, presence of lymphovascular space invasion, and positive peritoneal cytology. Six patients (9.8%) had ITCs in their SLN, while three (4%) were reported to have micrometastases. A prospective, multicenter cohort of patients with EC and low- to intermediate-risk preoperative features according to the ESGO/ESTRO/ESP guidelines [10] was investigated by Burg et al. [78]. Of 144 at least unilaterally mapped patients, 3 had ITCs in their lymph nodes (2%), and 7 had micrometastases (4.9%).

### 2.3. Prevalence of Low-Volume Metastases after Stratification of Patients in Intermediate- and High-Risk Groups

A multicenter study by Ducie et al. [76] reported the prevalence of micrometastases and ITCs in 202 patients with intermediate and high-risk EC who underwent SLN sampling. Intermediate risk was defined by the presence of endometrioid histology, any grade, and >50% myometrial invasion, while high risk included serous and clear cell tumors. Twenty patients had LVM in their lymph nodes. Twelve (5.9% of the total cohort) had ITCs, including seven intermediate-risk and five high-risk cases (8.5% and 4.2% of the total intermediate-risk and high-risk patients, respectively). Micrometastases were reported in eight cases (4% of the total cohort): six intermediate-risk and two high-risk cases (7.3% and 1.7% of the total intermediate-risk and high-risk patients, respectively).

In 2019, Persson et al. [33] described the prevalence of LVM in the SHREC study. They enrolled patients with high-risk EC (FIGO grade 3, non-endometrioid histology, >50% myometrial invasion, cervical stromal invasion, or non-diploid cytometry) in a prospective non-randomized trial to evaluate the accuracy of the SLN mapping algorithm. All patients underwent SLN mapping and full pelvic lymphadenectomy. Para-aortic lymphadenectomy was performed in the majority of patients (infrarenal para-aortic lymphadenectomy was performed in 80.9% of cases, and inframesenteric para-aortic lymphadenectomy in 3.5%). Nine patients were diagnosed with micrometastases (3.5%), while ten had ITCs (3.9%) in their SLN.

### 2.4. Prevalence of Low-Volume Metastases after Stratification of Patients in Low-, Intermediate-, and High-Risk Groups

In 2011, Ballester et al. [31] reported the incidence of LVM in a prospective cohort of 111 patients with FIGO stage I-II EC who had pelvic SLN assessment and subsequent systematic lymphadenectomy. They found that seven (6.3%) had LVM in their pelvic lymph nodes, including one (0.9%) who had ITCs, and eight patients (7.2%) were found to have macrometastases. They also reported the prevalence of SLN metastases after risk stratification. Among 57 low-risk patients (endometrioid EC, FIGO 2009 stage IA, grade 1 or 2), 6 (11%) had metastases in pelvic SLNs. Among intermediate-risk patients (stage IA, grade 3, or stage IB grade 1 or 2), the incidence of SLN metastases was 15.2% (five of 33 patients). While the high-risk patients (endometrioid, stage IB, grade 3, or non-endometrioid, any stage and grade) had the highest incidence of SLN metastases (5 of 16 patients, 31.3%); however, the type of metastasis was not further specified.

Kennard et al. [65] reported the prevalence of SLN metastasis stratified by risk class in a population of 414 patients who underwent robotic hysterectomy, SLN mapping, pelvic lymphadenectomy, and para-aortic lymphadenectomy directed by frozen section for apparent early-stage EC. Patients were classified as low-risk if they had endometrioid EC of any grade, FIGO stage IA; intermediate-risk if their disease had endometrioid histology of any grade, FIGO stage IB; and high-risk as any lesion with non-endometrioid histology. The low-risk group consisted of 275 patients: 23 had LVM, of which 16 had ITCs (8.4% and 5.8% of the low-risk cohort, respectively). In the intermediate-risk group (*n* = 80 patients), they reported LVM in 26 patients, 17 of whom had ITCs (32.5% and 21.3%, respectively). Finally, out of 59 patients in the high-risk group, only 3 patients had ITCs in their SLN (5.1%), while micrometastases were present in 5 patients (8.5%), for a total of 8 patients with LVM (13.6%). Interestingly, they also found that the presence of ITCs was associated with a higher risk of para-aortic node metastasis in all groups. In contrast, the presence of ITCs in the SLN was associated with non-SLN pelvic node metastasis only in patients with invasion of at least 50% of the total myometrial thickness.

In the study by Lavecchia et al., patients were further classified by risk cohort according to the ESGO/ESTRO/ESP guidelines [10]; the prevalence of LVM was 1.2% in the low-risk group (stage IA endometrioid, low-grade, no or focal LVSI), 2.6% in the intermediate-risk group (stage IB endometrioid, low-grade, no or focal LVSI or stage IA endometrioid, high-grade, no or focal LVSI or stage IA non-endometrioid without myometrial invasion), 0.8% in the intermediate-high-risk group (stage IA endometrioid with LVSI, or stage IB endometrioid, high-grade, or stage II disease), and 2.8% in the high-risk group (non-endometrioid disease with myometrial invasion, any stage, or stages III-IVA).

### 2.5. Considerations on the Prevalence of Low-Volume Metastases across Studies

The results reported above demonstrate that population characteristics impact the prevalence of LVM, either as micrometastases or ITCs. Table 1 shows that the prevalence of LVM can vary widely in an apparent early-stage EC population, although studies enrolling more than a thousand patients show that the prevalence of LVM ranges from 1.9% [59] to 10.2% [60]. Table 2 shows the prevalence of LVM in several cohorts stratified according to the patients’ risk of recurrent disease. Patients with low-risk EC, generally defined by the presence of endometrioid histology, low-grade, stage IA disease, according to NCCN guidelines [35] and ESGO/ESTRO/ESP guidelines [10], showed a low prevalence of LVM detected after ultrastaging and, specifically, the rate of positive nodes was less than 1% in the absence of myometrial invasion [60] (Table 2). In contrast, LVM were diagnosed in patients with intermediate- and high-risk EC at rates ranging from 7.4% [33] to 14.8% [74], and positive nodes were found in up to 10% of patients with non-invasive serous carcinoma [60] (Table 2). The clinical implications of these findings are discussed below.

## 3. Clinical Significance of Low-Volume Metastases

The ESGO/ESTRO/ESP guidelines, published in 2021, state that both macrometastases and micrometastases in the lymph nodes are considered metastasis and lead to upstaging [10]. The latest NCCN guidelines [35] confirm that ultrastaging is recommended when an SLN mapping technique is used to exclude the presence of LVM. The AJCC staging manual [58] recommends that the presence of ITCs is clearly registered even if they do not affect overall staging. This is also reported in the latest NCCN guidelines [35], the ESGO/ESTRO/ESP guidelines [10], and the most recently published FIGO staging system [6]. If ITCs are present, the stage would be pN0(i+). The new FIGO staging system [6], published in 2023, divides stage IIIC into stages IIIC1i and IIIC2i if micrometastases are present in the pelvic and/or para-aortic lymph nodes, and stages IIIC1ii and IIIC2ii if macrometastases are present in the pelvic and/or paraaortic lymph nodes (Figure 1, panel 4). This substaging is based on the concept that micrometastases confer a better prognosis compared to macrometastases [63,74,79].

The dimension of nodal metastases has an independent prognostic impact in patients with EC and positive lymph nodes [80]. The presence of macrometastases in the pelvic nodes is associated with the extent of pelvic involvement and with para-aortic nodal invasion [81]. On the other hand, unlike other types of nodal metastases, the presence of ITCs does not cause upstaging. Their prognostic role is still unclear, and there is no consensus on the ideal treatment, if any, or surveillance regimen to be offered to patients with ITCs in their pelvic nodes. Prospectively collected, conclusive evidence on the prognostic value of LVM in EC is still lacking. In a study by Todo et al. [74], ultrastaging was performed in 63 patients with intermediate-risk EC. They found that LVM were present in nine patients. Given the limited patient cohorts, they did not find a significant difference in overall survival (OS) and recurrence-free survival (RFS) (all *p* > 0.05) between the LVM and node-negative groups, although the 8-year OS/RFS rates were >20% lower in the LVM group than in the node-negative group (OS: 71.4% vs. 91.9%; RFS: 55.6% vs. 84.0%). They also reported that LVM was a significant predictor of extrapelvic recurrence, even after adjusting for other significant risk factors such as histologic grade, myometrial invasion, cervical invasion, lymphovascular space invasion, positive peritoneal cytology, and adjuvant treatment [adjusted risk ratio 17.9 (CI 95% 1.37–232.2)].

A retrospective study by St Clair et al. [82] included 844 patients with apparent early-stage EC and evaluated treatment patterns and oncologic outcomes in patients with LVM. A total of 44 patients had LVM (5.2% of the total cohort), including 23 patients with ITCs (2.7% of the total cohort). Most patients with ITCs, micrometastases, and macrometastases received adjuvant chemotherapy (83%, 81%, and 89%, respectively), while 106 (14%) of 753 node-negative patients received it. The 3-year RFS was 90% for patients with negative nodes, 86% for ITCs, 86% for micrometastases, and 71% for macrometastases (*p* < 0.001). In a subcohort of patients with endometrioid EC, 3-year RFS was 93% for patients with negative nodes, 94% for those with ITCs, 92% for those with micrometastases, and 85% for those with nodal macrometastases (*p* < 0.001). They concluded that patients with LVM have better oncologic outcomes when treated with surgery and adjuvant therapy than those with macrometastases.

In 2017, Plante et al. [63] published a prospective observational study of 519 patients who underwent surgery and SLN mapping for apparent early-stage EC. They aimed to compare the oncologic outcomes of patients with macrometastases, micrometastases, and ITCs in their SLN. They found that 85 patients (16.4%) had SLN metastases, of which 43 (51%) were macrometastasis, 11 (13%) micrometastasis, and 31 (36%) ITCs. The 3-year progression-free survival (PFS) was 95.5% in patients with ITCs. It was not significantly different from the 3-year PFS of patients with negative nodes (87.6%) or micrometastases in their SLN (85.5%). Patients with macrometastases had a significantly worse outcome with a 3-year PFS of 58.6% (*p* < 0.01 compared to ITC patients). Of the thirty-one patients that were diagnosed with ITCs, only one recurred. The characteristics of this patient’s disease classified her at high risk of recurrence, with non-endometrioid histology and deep myometrial invasion, and she received adjuvant treatment with chemotherapy and radiotherapy. In their report, none of the ITC patients who did not receive adjuvant treatment (10 patients) recurred. Therefore, the authors argue that appropriate adjuvant treatment should be proposed to patients with other risk factors, not those with ITCs and otherwise low-risk diseases.

In a report by Garcia Pineda et al. [67], including patients with FIGO stage I-II EC, 196 patients with negative nodes, 14 patients with nodal LVM, and 20 patients with macrometastases were compared. Patients with macrometastases showed significantly worse PFS than LVM and node-negative patients (61.1% vs. 71.4% vs. 83.2%; *p* = 0.018). The same findings were confirmed for OS: 50% vs. 78.6% vs. 81.5%, respectively (*p* < 0.001). Patients with LVM received adjuvant treatment in 71.5% of cases (28.6% had external beam radiotherapy with or without brachytherapy, and 42.9% had chemotherapy with or without external beam radiotherapy) and their outcome was not significantly different from those with node-negative disease.

To evaluate the impact of nodal micrometastases on oncologic outcomes, Ignatov et al. [83] compared two cohorts of patients with micrometastases—95 who received adjuvant treatment and 31 who did not—with a cohort of 302 node-negative patients who received no adjuvant therapy. When comparing patients without adjuvant treatment, the DFS rate was 32.3% and 73.2% for patients with and without micrometastases, respectively (*p* = 0.0001). On the other hand, the DFS was similar between node-negative patients without adjuvant treatment and patients with micrometastases who received adjuvant treatment (73.2% vs. 78.9%, *p* > 0.05). These findings were confirmed after adjusting for other significant risk factors. In this cohort of patients with micrometastases, adjuvant therapy (reported as radiation and/or chemotherapy) improved DFS and reduced the relative risk of recurrence by 71%.

In a study by Piedimonte et al. [84], twenty-three patients with low-grade endometrioid EC and LVM were compared with a group of EC patients matched for age, BMI, grade, depth of myometrial invasion and lymphovascular space invasion. The 1:1 matching was performed with propensity score analysis. Eleven patients had ITCs, while 12 had micrometastases in their lymph nodes. Their analysis showed that PFS rates within 5 years were similar between patients with LVM and node-negative patients. Adjuvant chemotherapy was administered to 70% of LVM patients, while 18.4% of node-negative patients received brachytherapy alone for local control. The authors conclude that patients with LVM in otherwise well-differentiated stage I EC treated with adjuvant therapy have similar outcomes to matched node-negative patients.

In a multicenter, retrospective study published in 2021, Backes et al. [85] reported the oncologic outcomes and recurrence rates of 175 women with endometrioid adenocarcinoma and ITCs only (patients with micro- or macrometastases in any nodes were excluded). No adjuvant treatment or vaginal brachytherapy was administered to 76 (43%) patients, while 21 (12%) received External Beam Radiotherapy (EBRT). Seventy-eight patients (45%) received chemotherapy with or without EBRT. The authors found that adjuvant treatment (either chemotherapy or EBRT) was not associated with RFS (chemotherapy: HR 0.63, 95% CI 0.11–3.52, and EBRT: HR 0.90, 95% CI 0.22–3.61, respectively) after controlling for FIGO stage, lymphovascular space invasion, and FIGO grade. No significant difference in extra-vaginal recurrence rates was found between patients with or without chemotherapy (5.2% vs. 3.8%, *p* = 0.68). Therefore, the study suggests that gynecologic oncologists should not decide whether to prescribe adjuvant treatment based on the presence of ITCs but instead based on other disease features.

Ghoniem et al. [86] reported the oncologic outcomes of 247 patients with LVM in a multi-institutional, retrospective study. Among them, 132 had ITCs, while 115 had micrometastasis. Overall, the 4-year RFS was 77.6% (95% CI, 70.2%–85.9%), with a recurrence rate of 15.4% (38 patients recurred: 17 with ITCs and 21 with micrometastases). Multivariate analysis showed that non-endometrioid histology, lymphovascular space invasion, and uterine serosal invasion were independent predictors of recurrence. The authors also showed that 18 ITC patients with grade 1 endometrioid EC without the above-mentioned risk factors had a low risk of recurrence, as only one patient recurred. In a multicenter study by Lavecchia et al. [59], 19 of 1012 patients had LVM. None of the patients with ITCs (12 patients, 1.2%) had a recurrence, while 1 patient out of 7 with micrometastasis recurred at 17 months. All patients with micrometastases and 58% of patients with ITCs (7 and 12 patients, respectively) received adjuvant treatment.

In 2023, Buda et al. [68] described a series of 1428 women with apparent early-stage endometrial cancer with negative lymph nodes, LVM, and macrometastases (1242, 76, and 110 patients, and 87%, 5.3%, and 7.7% of the total cohort, respectively). The 3-year disease-free survival (DFS) was 90.6%, 84.3%, and 58.5%, respectively (*p* < 0.001), showing that macrometastases have a significantly worse prognosis compared to negative nodes and LVM. The authors reported that the 3-year DFS of patients with LVM was not statistically different from the survival of patients with negative nodes. Adjuvant therapy was administered to 35.6% of patients with negative nodes, 84.2% with LVM, and 95.4% with macrometastases. After adjusting for other risk factors, lymphovascular space invasion was the only feature significantly associated with recurrence. Adjuvant treatment and the type of nodal metastasis were not significantly associated.

Cucinella et al. [87] reported the recurrence rates and oncologic outcomes of 494 patients from 15 centers with FIGO stage IA, low-grade endometrioid EC, who did not receive adjuvant treatment, of whom 452 (91.5%) were node-negative and 42 (8.5%) had ITCs in their SLN. The authors reported that 21 patients (4.3%) recurred within 5 years from surgery (5 patients with ITCs, 16 node-negative). Non-vaginal recurrence occurred in 15 patients (4 patients with ITCs, 11 node-negative). Regarding oncologic outcomes, ITC patients had poorer RFS (log-rank *p* < 0.01) compared to node-negative patients [85.1%; CI 95% 73.8%–98.2% versus 90.2%; CI 95% 84.9%–95.8%]. After excluding patients with lymphovascular space invasion (14 patients, 7 with ITCs and 7 with negative nodes), RFS was still worse for ITC patients compared to node-negative patients [89.5% CI 95% 78.8%–100% versus 91.1% CI 95% 86.0%–96.6%], although it did not reach significance (log-rank *p* = 0.05). Overall survival was not significantly different between ITCs and node-negative patients. This finding was also confirmed after excluding patients with lymphovascular space invasion.

A meta-analysis of eight studies by Gomez-Hidalgo et al. [88] found a higher relative risk of recurrence in patients with LVM. The overall relative risk of recurrence in patients with LVM was 1.34 [95% CI: 1.07–1.67]. They also reported the relative risk of recurrence between adjuvant-free, node-negative patients and LVM patients without adjuvant treatment [RR 2.26, 95% CI 0.44–11.70] and between adjuvant-free LVM patients and LVM patients who received adjuvant therapy [RR 1.05, 95% CI 0.83–1.34]. Regardless of adjuvant treatment, the relative risk of recurrence was not statistically different in these groups; thus, no strong indication could be made regarding adjuvant therapy. Table 3 reports the recurrence rates of patients with EC and negative nodes, micrometastases, or ITCs. The reported risk of recurrence varies significantly among studies, and data on patients with ITCs are unfit for direct comparison given the different management patterns (specifically, administration of adjuvant treatment) of ITC patients across institutions. In fact, in recent years, the presence of LVM has influenced the rates and types of adjuvant treatment given to patients.

It remains controversial whether adjuvant therapy can modify the risk of recurrence and disease-free survival in patients with EC and LVM, especially when administered in the context of LVM in otherwise low- or intermediate-risk diseases. On the other hand, the more frequent presence of LVM in intermediate-high and high-risk patients hinders the understanding of the intrinsic impact of LVM on survival outcomes. Currently, the independent risk of recurrence associated with LVM in an otherwise low-risk population that has not received adjuvant treatment is unknown.

## 4. Present and Future Challenges

Surgical staging, including retroperitoneal staging, was introduced for EC in 1988 with the publication of the updated FIGO staging system [12]. Following the publication of two independent trials [17,18] showing no evidence of prognostic benefit for lymphadenectomy and the exploration of the sentinel node technique (which had already become standard in other surgical fields), sentinel node mapping and ultrastaging became the standard procedure for surgical staging of EC. While the incidence of LVM has been extensively described in several different patient cohorts and studies enrolling significant numbers of patients, the clinical significance of LVM and, specifically, the impact of ITCs on prognosis have to be confirmed in prospective studies. Patient counseling should highlight the limited knowledge and lack of prospectively collected evidence on ITCs. Recommendations on surveillance of patients with ITCs and otherwise low-risk EC who do not undergo adjuvant treatment are still lacking. An institutional agreement suggested clinical and radiologic follow-up every six months to detect recurrences early [89]. As advocated by Bogani et al. [79], prospective trials are needed to address this issue. To better understand the clinical significance of LVM, an observational study by Martinelli et al. began collecting data on LVM in endometrial and cervical cancer in 2020 (ITCMicroUtCa—NCT04403867), with an estimated completion date of early 2027. The study estimates 500 patients with endometrial or cervical cancer who underwent an SLN procedure and were diagnosed with nodal metastasis. Pathological analysis of lymph nodes will be performed with standard hematoxylin–eosin plus ultrastaging or OSNA assays. Primary outcome measures will be survival and recurrence rates, while the use of adjuvant treatment will be recorded as a secondary outcome. International prospective studies focused on patients with ITCs and promoted by Mayo Clinic are also expected to begin enrollment in the upcoming years.

On the other hand, the new molecular classification proposed by TCGA was shown to have an important prognostic role in predicting recurrence [5]. Molecular classification has not been adopted widely due to the high cost of the testing and limited prospective evidence to guide treatment decisions. With recent evidence that biological and molecular characteristics of the disease may predict prognosis independently of stage and histopathologic features and the introduction of molecular features in the 2023 FIGO staging [6], the role of surgical (and nodal) staging has been questioned. However, merging molecular features with nodal status and other traditional histopathologic features [90] rather than eliminating retroperitoneal staging may result in improved risk stratification. Ongoing observational studies, like the SENECA study (NCT05707312), are evaluating the rate of lymph node metastases in patients with apparent early-stage, molecularly classified EC undergoing primary surgery and an SLN procedure.

## 5. Conclusions

The prevalence of LVM after sentinel node mapping and ultrastaging has been reported to range from 1.9% to 10.2% in large cohorts of over one thousand patients with endometrial cancer, although it can vary significantly depending on disease characteristics. The impact of LVM, particularly ITCs, on recurrence risk and patient survival has yet to be determined. Prospectively collected data from larger, international cohorts will help to define the clinical significance of LVM in sentinel nodes and its potential influence on patient counseling and adjuvant treatment.

## Figures and Tables

**Figure 1 cancers-16-01338-f001:**
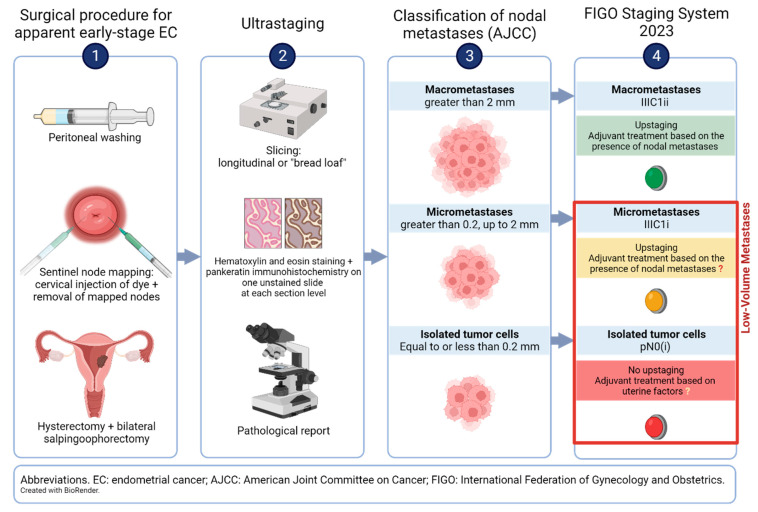
Surgical procedures, ultrastaging, pathological report, and current classification of nodal metastases in apparent early-stage EC.

**Table 3 cancers-16-01338-t003:** Recurrence rates in patients with negative nodes, micrometastases, and isolated tumor cells.

First Author	Number of Patients	Recurrences in Patients with Negative Nodes (%)	Recurrences in Patients with MM (%)	Recurrences in Patients with ITCs (%)	Non-Vaginal Recurrences in Patients with ITCs (%)	Recurrences in Patients with ITCs and Otherwise Low-Risk EC	Follow-Up (Months)
Kim [49]	508	NR	NR	2 (10.5)	2 (10.5)	0	NR
Raimond [73]	136	11 (9.7)	1 (6.7)	NR	NR	NR	NR
Todo [74]	61	8 (15)	1 (33.3)	3 (50)	3 (50)	0	107
St Clair [82]	844	47 (6)	2 (9.5)	2 (8.7)	1 (4.3)	NR	26
Plante [63]	519	NR	NR	1 (3.2)	1 (3.2)	0	29
Backes [64]	184	NR	1 (50)	0	0	0	31
Garcia Pineda [67]	230	NR	2 (25)	3 (33.3)	0	1	60
Lavecchia [59]	1012	44 (4.6)	1 (14.3)	0	0	0	27
Buda [68]	1428	75 (5.9)	6 (12)	5 (19.2)	NR	NR	33.3
Backes [85]	175	16 (3.5)	NR	9 (5.1)	8 (4.6)	1	31
Ghoniem [86]	247	NR	21 (18.3)	17 (12.9)	NR	1 (of 18 with low-risk EC)	29.6
Cucinella [87]	494	16 (3.5)	NR	5 (11.9)	4 (9.5)	Only low-risk patients included	28 in ITCs and 31 in node-negative

Abbreviations: EC: endometrial carcinoma; MM: micrometastases; ITCs: isolated tumor cells; NR: not reported.

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
