# Peer review of "Low-Volume Metastases in Apparent Early-Stage Endometrial Cancer: Prevalence, Clinical Significance, and Future Perspectives"

_cancers, 2024, doi:10.3390/cancers16071338_

Round 1

Reviewer 1 Report

Comments and Suggestions for Authors

Authors present an interesting review on the prevalence and clinical significance of LVM detected in the sentinel nodes during EC staging. The manuscript is well writtent is worth of publication.

I have only a few sugestions to improve the overall quality:

- TCGA molecular subgroups of endometrial cancer should be discussed in deep in the first paragraphs.

- A separate paragraph regarding iintraoperative examination of sentinel nodes should be included (accuracy of frozen section, pitfalls, utility of frozen section vs permanent sections)

- Sentinel nodes ultrastaging protocols should be discussed in deep: please discuss the main protocols: MD Anderson Cancer Center  and  Memorial Sloan-Kettering Cancer Center (include references such as PMID: 35280827)

- A brief paragraph on the utility of One-Step Nucleic Acid Amplification (OSNA) in sentinel nodes of EC patients should be included.

Author Response

Thank you for your feedback and suggestions for improvement. Please see our responses (in blue) to your suggestions below: 

- TCGA molecular subgroups of endometrial cancer should be discussed in deep in the first paragraphs. Response: the introduction and discussion on TCGA molecular subgroups have been moved to lines 68-79 instead of 474 and following lines: In addition to traditional histopathologic findings and disease spread, The Cancer Genome Atlas (TCGA)[5] identified four molecular EC subgroups with distinct prognostic characteristics: copy-number high with poor prognosis, copy-number low and mismatch-repair deficient with intermediate prognosis and POLE-mutated with an overall good prognosis. This classification relied on expensive analyses on fresh-frozen samples, but new systems based on the same categories as identified by TCGA – like the Proactive Molecular Risk Classifier for Endometrial Cancer (ProMisE) – proposed more pragmatic methods that can be leveraged in clinical practice[8,9]. The updated molecular classification includes four groups: p53-abnormal (instead of copy-number high), non-specific molecular profile (instead of copy-number low), mismatch-repair deficient and POLE-mutated [9]. This classification has been included in the latest ESGO/ESTRO/ESP guidelines [10] and 2023 FIGO staging [6].  

- A separate paragraph regarding intraoperative examination of sentinel nodes should be included (accuracy of frozen section, pitfalls, utility of frozen section vs permanent sections) Response: please see lines 169-179 of the updated manuscript. Ultrastaging is performed after permanent sectioning of the SLN. A potential limitation of SLN ultrastaging on permanent sections, without frozen section analysis, is that in around 8% of cases an empty node is found (i.e., a specimen that contains only fibroadipose and no lymphoid tissue at the pathologic evaluation)[53]. Casarin et al. proposed frozen section as a method to determine the accuracy of the SLN bioptic procedure, ensuring an adequate nodal staging [53]. Bellaminutti et al.[54] also reported that the accuracy of frozen section in detecting nodal metastases is 93%, with a positive predictive value of 100%. A systematic review and meta-analysis[55] also reported that intraoperative SLN analysis on frozen section increased the SLN detection rates. However, the ESGO/ESTRO/ESP guidelines state that frozen section analysis should not replace final pathologic examination and ultrastaging[10].  

- Sentinel nodes ultrastaging protocols should be discussed in deep: please discuss the main protocols: MD Anderson Cancer Center and Memorial Sloan-Kettering Cancer Center (include references such as PMID: 35280827) Response: the ultrastaging section has been improved, see lines 155-167. The Memorial Sloan Kettering SLN processing protocol includes 1) hematoxylin/eosin staining, 2) if standard examination is negative, from each paraffin block two adjacent 5 µm sections at each of 2 levels, 50 µm apart, are cut longitudinally, and 3) at each level one slide is stained with hematoxylin and eosin and one with immunohistochemistry using anticytokeratin AE1:AE3 for a total of 5 slides per block[50]. At MD Anderson,  two different protocols - an intensive method and an abbreviated version - have been compared[52] and no difference in detecting nodal metastases was found. In the first protocol, 5 levels are stained with hematoxylin and eosin at 250 µm intervals, with 2 unstained slides at each level; then, pankeratin immunohistochemistry is performed on one slide in cases with negative hematoxylin and eosin examination. The second protocol involves staining with hematoxylin/eosin only on one level instead of 5, with two accompanying unstained slides. The ultrastaging steps are shown schematically in Figure 1, panel 2. 

- A brief paragraph on the utility of One-Step Nucleic Acid Amplification (OSNA) in sentinel nodes of EC patients should be included. Response: Please see lines 180-205 of the updated manuscript. Recently, new molecular techniques have been explored to detect nodal metastases in SLN. Nagai et al.[56] first reported on the effectiveness of a one-step nucleic acid amplification (OSNA) assay using cytokeratin 19 (CK19) mRNA for diagnosing LN metastasis in EC patients. Normal lymphoid tissue does not express CK19, but cancer cells and their metastases do. The Authors found that the new diagnostic method had the same diagnostic capability as the frozen-section histology, with high concordance between the two methods (97.1%). The technique was later applied to 135 SLN (58 patients) and compared with ultrastaging, leading to the upstaging of 20.7% of cases, in a comparative study by Kost’un et al.[57]. They reported a sensitivity of the OSNA assay of 90.9%, a specificity of 85.5% and concordance of 85.9% with ultrastaging. In 2023, a multicenter prospective study on the detection of nodal metastases in cervical and endometrial cancer patients, conducted in six institutions in Japan,[58] showed that the diagnostic ability of the OSNA assay in detecting metastases was equivalent to that of histopathological examination. Specifically, the Authors reported a concordance rate of 97.9% (95% CI: 96.1–99.1%), with a sensitivity and specificity of the OSNA assay of 91.8% (95% CI: 81.9–97.3%) and 98.9% (95% CI: 97.3–99.7%), respectively. Each lymph node (of 437 samples retrieved from 133 endometrial and cervical cancer patients who underwent systematic lymphadenectomy) was cut with 2-mm intervals along the short axis direction, and alternate sections were used for the OSNA assay and standard hematoxylin-eosin examination. Ultrastaging with anti-CK19 antibody immunostaining was not routinely performed. This procedure was done in cases of non-concordance of the standard pathological examination and OSNA assay. In conclusion, the OSNA assay could be a promising new method to diagnose nodal metastases in patients with EC. However, its cost-effectiveness compared to standard histopathologic examination and CK19 immunostaining, as well as broad applicability in standard clinical practice, have not yet been assessed.

Reviewer 2 Report

Comments and Suggestions for Authors

General comment: The manuscript provides a narrative review focusing on the evolving and critical topic of clinical management of patients with endometrial carcinoma, specifically addressing the presence of micrometastases or isolated tumor cells found during the pathological evaluation of lymph nodes. The text is well-organized and articulately and presents current knowledge and practices in the treatment of endometrial tumors, with clear tables and a high-quality figure. The conclusions drawn are appropriate and reflect a thorough review of existing literature, albeit without particularly novel insights or systematic analysis. Nonetheless, the manuscript is a valuable resource for professionals involved in treating gynecological tumors, offering a substantial update on the topic.

Organization and Articulation: The manuscript is commendably organized and articulated, presenting complex information in an accessible manner. Each section logically flows into the next, allowing readers to be informed on clinical management strategies for endometrial carcinoma with micrometastases or ITC.

Literature Review: The review of the literature is comprehensive regarding covering the topic. However, it lacks particular originality and systematicity, which might have added more depth to the analysis. Despite this, it serves as a potentially useful tool for those seeking an in-depth update on managing patients of this setting.

Specific comment: Adding a paragraph that discusses emerging research with more details could also enrich the manuscript, providing readers with insight into potential future directions in the field.

Author Response

Thanks for your feedback and comment on the manuscript. Please find our response to your specific comment below in blue.  

Specific comment: Adding a paragraph that discusses emerging research with more details could also enrich the manuscript, providing readers with insight into potential future directions in the field. Emerging research has been outlined in the paragraph “Present and future challenges” and has been further explored in the revised paragraph on ultrastaging by adding a section on the one-step nucleic acid amplification (OSNA) for detecting sentinel lymph node metastasis. 

Reviewer 3 Report

Comments and Suggestions for Authors

The paper titled "Low-volume metastases in apparent early-stage endometrial cancer: Prevalence, clinical significance, and future perspective" offers an interesting review regarding a much-discussed topic in the surgery of endometrial cancer.

The abstract is well written (line 40 please take out the comma) but slightly long, potentially hindering its ability to effectively communicate the main findings briefly. 

I have a few suggestions to make this study more complete by including some of the topics that were omitted:

The paper omits any mention of histological subtypes as independent parameters of evaluation. A more comprehensive analysis of the literature, including the more aggressive subtypes, such as uterine serous carcinoma, would be beneficial for this paper's objective.

Also, the current paper lacks reference to the potential use of one-step nucleic acid amplification (OSNA) for detecting sentinel lymph node metastasis. The OSNA assay, known for its high sensitivity and specificity, presents itself as a promising molecular tool for the detection of metastasis in patients with endometrial carcinoma. The incorporation of such alternative approaches in the discussion could further enrich the paper's scope and offer readers a more comprehensive perspective on available diagnostic tools.

In summary, while the paper provides valuable insights into low-volume metastases in early-stage endometrial cancer, addressing the concerns related to the abstract length, including histological subtypes as independent factors, and acknowledging alternative diagnostic tools like OSNA would enhance the overall robustness and applicability of the study.

Author Response

Thank you for your feedback and suggestions. Please find a point-by-point response below (text in blue) and the edits in the attached manuscript (track changes ad comments).  

The abstract is well written (line 40 please take out the comma) but slightly long, potentially hindering its ability to effectively communicate the main findings briefly. Please find in the latest version of the manuscript an edited version of the abstract (294 words instead of 338). 

The paper omits any mention of histological subtypes as independent parameters of evaluation. A more comprehensive analysis of the literature, including the more aggressive subtypes, such as uterine serous carcinoma, would be beneficial for this paper's objective. Please see the following edit (lines 59-68 of the updated manuscript): The anatomical extent of the disease, including metastases to the lymph nodes, is an important prognostic factor. In addition, the histopathological subtype[4] and the molecular features of the disease[5] have also been described as independent predictors of prognosis. In the 2023 FIGO staging the presence of an aggressive histopathological subtype leads to upstaging also when the disease is confined to the uterus if there is myometrial invasion [6]. Aggressive histopathological subtypes include high-grade endometrioid cancers, serous, clear cell, undifferentiated, mixed, mesonephric-like, gastrointestinal mucinous type carcinomas, and carcinosarcomas[6]; it is well known that in non-endometrioid disease survival is significantly reduced[7]. 

Also, the current paper lacks reference to the potential use of one-step nucleic acid amplification (OSNA) for detecting sentinel lymph node metastasis. The OSNA assay, known for its high sensitivity and specificity, presents itself as a promising molecular tool for the detection of metastasis in patients with endometrial carcinoma. The incorporation of such alternative approaches in the discussion could further enrich the paper's scope and offer readers a more comprehensive perspective on available diagnostic tools. Please see the following edits (lines 180-205). Recently, new molecular techniques have been explored to detect nodal metastases in SLN. Nagai et al.[56] first reported on the effectiveness of a one-step nucleic acid amplification (OSNA) assay using cytokeratin 19 (CK19) mRNA for diagnosing LN metastasis in EC patients. Normal lymphoid tissue does not express CK19, but cancer cells and their metastases do. The Authors found that the new diagnostic method had the same diagnostic capability as the frozen-section histology, with high concordance between the two methods (97.1%). The technique was later applied to 135 SLN (58 patients) and compared with ultrastaging, leading to the upstaging of 20.7% of cases, in a comparative study by Kost’un et al.[57]. They reported a sensitivity of the OSNA assay of 90.9%, a specificity of 85.5% and concordance of 85.9% with ultrastaging. In 2023, a multicenter prospective study on the detection of nodal metastases in cervical and endometrial cancer patients, conducted in six institutions in Japan,[58] showed that the diagnostic ability of the OSNA assay in detecting metastases was equivalent to that of histopathological examination. Specifically, the Authors reported a concordance rate of 97.9% (95% CI: 96.1–99.1%), with a sensitivity and specificity of the OSNA assay of 91.8% (95% CI: 81.9–97.3%) and 98.9% (95% CI: 97.3–99.7%), respectively. Each lymph node (of 437 samples retrieved from 133 endometrial and cervical cancer patients who underwent systematic lymphadenectomy) was cut with 2-mm intervals along the short axis direction, and alternate sections were used for the OSNA assay and standard hematoxylin-eosin examination. Ultrastaging with anti-CK19 antibody immunostaining was not routinely performed. This procedure was done in cases of non-concordance of the standard pathological examination and OSNA assay. In conclusion, the OSNA assay could be a promising new method to diagnose nodal metastases in patients with EC. However, its cost-effectiveness compared to standard histopathologic examination and CK19 immunostaining, as well as broad applicability in standard clinical practice, have not yet been assessed.